# Key risk factors for substance use among female sex workers in Soweto and Klerksdorp, South Africa: A cross-sectional study

Ellis Jaewon Yeo[1]*, Khuthadzo Hlongwane[2], Kennedy Otwombe[2,3], Kathryn L. Hopkins[4], Ebrahim Variava[3], Neil Martinson[2,5,6], Steffanie A. Strathdee[7,8], Jenny Coetzee[2,9,10], Minja Milovanovic[2,10]

1 Harvard Global Health Institute, Harvard University, Cambridge, Massachusetts, United States of America, 2 Perinatal HIV Research Unit (PHRU), Faculty of Health Sciences, University of The Witwatersrand, Soweto, South Africa, 3 School of Public Health, Faculty of Health Sciences, University of The Witwatersrand, Johannesburg, South Africa, 4 Sabin Vaccine Institute, Washington, DC, United States of America, 5 Department of Internal Medicine, Klerksdorp Tshepong Hospital Complex, University of The Witwatersrand, Matlosana, South Africa, 6 Johns Hopkins University Center for TB Research, Baltimore, Maryland, United States of America, 7 Department of Medicine, University of California San Diego, San Diego, California, United States of America, 8 Johns Hopkins Bloomberg School of Public Health, Johns Hopkins University, Baltimore, Maryland, United States of America, 9 South African Medical Research Council, Cape Town, South Africa, 10 African Potential Management Consultancy, Kyalami, South Africa

* ellisjyeo@gmail.com

## Abstract

### Introduction

Sex workers in South Africa experience high levels of trauma and mental health issues, but little is known about their drug and alcohol use. This study assessed the prevalence of substance use and its key risk factors amongst female sex workers (FSWs) at two sites in South Africa.

### Methods

Two cross-sectional studies were conducted, in Soweto and Klerksdorp, South Africa. Using respondent-driven sampling (RDS) 508 FSWs in Soweto and 156 in Klerksdorp were enrolled. A study-specific survey was used to collect social and demographic information, substance use, mental ill-health, and HIV status. Raw and RDS-adjusted data were analyzed using Chi-squared tests of association. Weighted and unweighted Poisson regression models were used to assess key risk factors for alcohol and drug use at both univariate and multivariate levels.

### Results

Of the 664 FSWs, 56.2% were binge drinkers and 29.4% reported using drugs within the last year. Living in a home with regular food (RR: 1.2597, 95% CI: 1.1009–1.4413) and being HIV positive (RR: 1.1678, 95% CI: 1.0227–1.3334) were associated with a higher risk of binge drinking. Having symptoms suggestive of post-traumatic stress disorder (RR: 1.1803, 95% CI: 1.0025–1.3895) and past year physical/sexual abuse from either intimate

**Data Availability Statement:** Data cannot be shared publicly because female sex workers are a highly vulnerable key population due to violence,

stigma and the criminalised nature of the work. For this reason it is imperative that data collected from studies with female sex workers follow ethical considerations and account for the sensitive nature of the information. Datasets will be available upon request by contacting info@phru.co.za.

**Funding:** The Soweto study was funded by the Medical Research Council of South Africa in terms of the National Health Scholars Programme from funds provided for the purpose of a PhD by the National Department of Health/Public Health Enhancement Fund to JC, https://www.samrc.ac.za/researchdevelopment. In addition, funding was received through The Albert Wessels' Trust to JC. The Global Fund and Networking HIV/AIDS Community of South Africa (NACOSA) provided support to the project, NAC-SW-2016-1, awarded to JC. The Klerksdorp study was funded by UC San Diego Center for AIDS Research International Pilot Grant Application to NM and JC, (2P30AI036214-24). For the purpose of open access, the author has applied a CC BY public copyright license to any Author Accepted Manuscript version arising from this submission. The funders had no role in study design, data collection and analysis, decision to publish, or preparation of the manuscript.

**Competing interests:** The authors have declared that no competing interests exist.

(RR: 1.3648, 95% CI: 1.1522–1.6167) or non-intimate partners (RR: 1.3910, 95% CI: 1.1793–1.6407) were associated with a higher risk of drug use.

## Discussion

In conclusion, our findings demonstrate a high prevalence of alcohol and drug use among FSWs in Soweto and Klerksdorp with site-specific contextual dynamics driving substance use. Site differences highlight the importance of tailoring site-specific substance use harm mitigation for this key population.

## Introduction

South Africa has one of the highest levels of alcohol consumption per adult drinker, globally [1]. While women generally drink less alcohol than men [2], female sex workers (FSWs) have been found to have a greater likelihood of a past year diagnosis of substance use disorder compared to women who do not sell sex [3]. Based on a review of global literature, 19–76.5% of FSWs use alcohol prior to and during sex work. The large range demonstrates that levels of alcohol use vary widely, and this is likely a consequence of cultural and other environmental factors [4].

A high proportion of FSWs in South Africa have been described as engaging in hazardous alcohol consumption, defined as scoring 3 or higher on the AUDIT-C scale of 0–12 [5]. For example, 81.5% of FSW in Johannesburg, 58.4% in Cape Town, 43.0% in Durban [6], and 84.8% in Soweto [7] were classified as hazardous drinkers. There are no previously reported rates of binge drinking available specifically for the South African FSW population, though a Kenyan study reports that 33% of the study's FSW population were binge drinkers [8]. In addition to alcohol, studies have shown a high prevalence of substance use (alcohol and other drug use) amongst sex workers, both globally and in South Africa [9, 10]. A 2015 study of FSWs in KwaZulu-Natal found that 83.2% of respondents admitted to lifetime substance use [11]. Across the country, the use of opioids such as heroin, codeine, and Nyaope (antiretroviral therapy [ART] medication mixed with detergent, rat poison, marijuana, and/or methamphetamine) has been increasing [12, 13]. Alcohol and other drugs have been widely used by both FSWs and their male clients to facilitate participation in sex work [3, 4, 9]. Evidence suggests that taking substances may act as a coping strategy for stress resulting from stigma, violence, and/or emotional trauma [4, 14, 15].

In South Africa, the lifetime prevalence of substance use disorders among adults is 13.3%, and yet less than 5% ever receive treatment [16]. Women are significantly underrepresented in substance use treatment facilities, comprising only 10–24% of treatment centre patients across all nine South African provinces [17, 18]. One study of women in Pretoria showed that less than 20% of participants had any awareness of existing alcohol and drug treatment programmes [3]. There are no substance treatment centres dedicated to FSWs despite the high prevalence of substance use in this population, and access to the few treatment centres is limited, partially due to the limited availability of space and services [3, 19, 20]. Most treatment centres are unable to address gender and work-specific needs of FSW, such as childcare, trauma, and competing financial priorities, since these women cannot earn an income while in treatment. These barriers make services even more inaccessible to FSW most at need for support [3, 21, 22]. FSWs often face discrimination in public facilities due to the criminalisation of sex work [7, 13], which consequently reduces their access of health care.

FSWs have been found to experience overall higher levels of childhood trauma according to the childhood trauma questionnaire [23, 24] and of recent exposure to violence in comparison to the general population [25, 26]. A 2015 study assessing polyvictimisation of FSWs in Soweto, South Africa found higher levels of childhood trauma amongst FSWs compared to the general population, and this vulnerability continued into adulthood (25). Almost 86.6% of FSWs reported experiencing some form of violence in their lifetime [25]. The increased prevalence of substance use that FSWs face may account for the higher prevalence of mental health issues compared to the general population [11]. Findings from previous research highlight a high prevalence of depressive symptoms amongst FSWs, with more than two thirds of FSWs in Soweto experiencing symptoms suggestive of severe depression, and more than a third suggestive of post-traumatic stress disorder (PTSD) [7].

Mental health disorders and exposure to violence have both been associated with substance use disorder [27, 28]. Excessive drinking is a maladaptive coping strategy, and can be used to mask underlying mental health concerns such as depression and PTSD [4, 29]. High levels of alcohol consumption are a contributing factor for risky sexual behaviours and HIV acquisition among women in South Africa [30]. Furthermore, depressive symptoms, violence, and substance use are all associated with a poor adherence to ART among people living with HIV infection [11, 31, 32]. Given the goal to achieving the United Nations 95:95:95 targets by 2030 [33], and in a key population where HIV prevalence has been recorded to range between 34–90%, factors influencing poor adherence to ART can be problematic and directly impact viral suppression [6, 24, 34, 35].

Despite the high prevalence of trauma and mental health concerns reported by FSWs in South Africa, gaps exist when focusing on patterns of substance use for this population. We describe the prevalence of self-reported substance use (i.e.; binge drinking and drug use) and associated risk factors among FSW populations from Soweto and Klerksdorp, South Africa.

## Methods

### Study design, setting and sampling

Two separate cross-sectional studies were conducted, first in Soweto and then in Klerksdorp, South Africa. The two studies used the same respondent driven sampling (RDS) methodology, where an initial respondent is recruited based on network size, issued with coupons and asked to invite additional participants from within their network. Respondent driven sampling is frequently used to enroll hard to reach populations for population-level estimates [36, 37]. In Soweto, 508 FSW were recruited between February and September 2016 [38]. The methodology from the Soweto study has been thoroughly described in previous publications [7]. A replica RDS study was undertaken in Klerksdorp in 2018, enrolling 156 FSW. The sample sizes across both studies were based on an estimated number of FSW for each geolocation alongside respective HIV prevalence estimates [7].

Soweto is a predominantly urban and peri-urban, low-income township on the outskirts of Johannesburg, South Africa. It has the highest population density in South Africa, with over a million inhabitants [39]. Soweto has approximately 3,000 drinking establishments (legal and illegal), with over R50 million (USD 3.8 million) spent annually on beer [24]. Sex work is often undertaken informally within drinking establishments, in private homes, and in open spaces, with few formal brothel or strip-club type venues. While FSW report having between 0–19 clients per day, informal sex work was also paid for in beer [24]. Klerksdorp is a gold and platinum mining town in the Matlosana Municipality in the Dr Kenneth Kaunda district, North West Province, made up of peri-urban townships. The district is home to almost two hundred

thousand inhabitants. Sex work is common within mining and trucking communities [40–42], however little was known about this population of FSW.

Accessing FSW populations poses unique and often challenging ethical concerns. Sex workers in South Africa are vulnerable due to the stigma associated with the work, the high levels of violence, and ongoing criminalisation of sex work. For this reason it is imperative to ensure the protection of participants from any harm (whether it be physical, emotional, legal, social or psychological) during the research. For both studies, data collection teams worked closely with local sex work program implementing partners to ensure that access to the sex work population was rapid, trust could be quickly earned, and effective linkage to care and follow-up for health or human rights concerns could be addressed. Sex work programs have been strategically designed to provide and facilitate rights-based access to healthcare as well as psychosocial and risk reduction services catering specifically to the needs of sex workers. By utilizing peer educators and an outreach model, the programs provide HIV Testing Services (HTS) and linkage to care, and widespread condom and lubricant distribution and are considered as pillars of public health care in the sex work sector.

## Study participants inclusion criteria

Across both sites, eligible participants were cisgender female, aged 18 years or older, sold sex within the past 6 months in the respective district, possessed a study specific and valid recruitment coupon (see below section), and gave voluntary informed consent to participate.

## Recruitment, screening and enrollment

A total of 11 seeds (initial participants) in Soweto and two seeds in Klerksdorp were used to initiate recruitment at each location. Seeds were identified by other FSWs as well-networked FSWs within their community during their monthly workshop sessions in the local sex work program. Seeds were selected based on the size of their network of FSWs in the sub-district (other FSWs they knew and had seen within the preceding two weeks). Similar to a chain referral method [36], all participants (including seeds) were given a maximum of three coupons with which to recruit additional potential participants. Seeds and subsequently enrolled participants were asked to give the coupons at random to women they knew and who knew them, who (like themselves) sold sex in the recruitment geolocation and who were 18 years or older. Potential participants with coupons arrived at the two sites for screening and enrolment procedures, including written informed consent to participate in the study, administered by sex worker peer educators in either English, isiZulu, Sesotho or Setswana.

## Data collection and management

Post enrollment, a 45-minute, interviewer-administered questionnaire was completed. Interviews were conducted in a private location and data was collected in real time onto tablets using the REDCap electronic data management system [43]. REDCap databases were designed with built-in skip patterns and algorithms and were monitored daily to ensure data quality. Participants were reimbursed for their time (250 ZAR; $19 USD), and a secondary incentive (20 ZAR; $1.5 USD) for successful chain-recruitment was provided 7–10 days later. Data were captured directly onto tablets using the REDCap electronic data management system [43], with built-in skip patterns and algorithms and monitored daily to ensure data quality. RDS assumptions [36, 37] were monitored during data collection, using specialist software (RDS-Analyst) [44].

## Questionnaire and study measures

The initial questionnaire was developed for the Soweto study using a community centric approach, and subsequently used for the Klerksdorp study (S1 Questionnaire). In adapting to the findings of the Soweto study and Klerksdorp contextual factors, the Center for Epidemiologic Studies Depression Scale (CES-D) short depression scale was used [45]. The questionnaire is comprised of a number of tools that have been previously validated (S1 Table). Workshops with FSW and peer educators at both sites were undertaken to obtain input and feedback on the study design and questionnaire. Cognitive interviews, to assess whether the questionnaire was understandable, appropriate, and colloquially suitable were conducted with FSW to improve the reliability of the questionnaire.

**Socio-demographics.**   The socio-demographic data collected included age; migration status (local vs internal [South African] immigrant vs external [cross-border] immigrant based on birthplace); highest level of education; food security (do people in your home go regularly without food?); and sexual history, including age of sexual debut, circumstances of first sex (coercive: 'I was tricked/forced/raped' vs non-coercive: 'I was willing/was persuaded'), and age first sold sex.

**HIV status and treatment.**   HIV status was determined by two concurrent rapid tests (Abon™ and First Response™). HIV positive participants self-reported ART use (never taken treatment vs on treatment always vs stopped taking treatment [defaulted]). Participants on ART self-reported their adherence, with adherence defined as either always being on treatment or having taken treatment between 5–7 days in the past week and non-adherence defined as having taken treatment less than 5 days in the past week. South Africa's ART policy had evolved numerous times prior and during the two study enrolment periods. At the time that the Soweto study was enrolling participants, adults with a CD4 count of <500 cells/mm$^3$ were eligible to initiate treatment [46]. However, universal test and treat (UTT) was introduced in South Africa on 1 September 2016, thus making ART available to all HIV infected persons regardless of CD4 count [47] and same day ART initiation came into effect from 1 September 2017 [48].

**Mental health, experienced trauma, and substance use.**   Depression was measured using the 10-item CES-D short scale [45]. Depression scores were calculated and a cut-off of 9 was used to indicate major depressive symptoms. Questions included 'During the past week I was worried by things that usually don't worry me', and 'During the past week I felt I was just as good as other people'. Responses ranged from: 0 'rarely/none of the time', 1 'some of the time (1-2days)', 2 'a moderate amount of time (2–4 days)' and 3 'most of the time (5–7 days)'. Scores were tallied and the overall scale alpha was 0.68.

Post-traumatic stress disorder was measured using the PTSD-8 scale [49]. Questions were categorised into three subscales in line with the DSM-IV [49] (hypervigilance, intrusion and avoidance). Participants were asked 8 questions referring to having 'recurrent thoughts or memories of the event', and 'feeling jumpy, get a fright easily', when they thought about any event which they had found traumatic. If participants showed signs of hypervigilance, intrusion, or avoidance then they were considered to have some PTSD symptoms. Scores were summed and the overall Cronbach alpha was 0.90.

The Childhood Trauma Questionnaire [50] measures 4 dimensions of use: neglect, emotional, physical and sexual abuse. Items for these dimensions were scored separately and if participants had any sign of use from at least one of the dimensions, it indicated some childhood trauma. The overall Cronbach alpha for the summed scores was 0.78.

Exposure to violence was assessed using the WHO violence questionnaire (adapted for female sex workers) [51].

Physical abuse by intimate partner or non-intimate partner (client, police, family and other men) was assessed using the following items: "Within the past year did any partner slap you, push you or throw something at you which could hurt you?", "Within the past year did any partner hit you with a fist or with something else (such as a beer bottle, stick or belt) which could hurt you?", "Within the past year did any partner kick, drag, beat, choke or burn you?", and "Within the past year did any partner threaten to use or actually use a gun, knife or other weapon against you?". Two variables were created using these items (physical abuse by intimate partner and physical abuse by non-intimate partner) with responses being categorised into none vs. some if participants had experienced any of the use from the items.

Sexual abuse by intimate partner or non-intimate partner (client, police, family and other men) was determined using the following items: "Within the past year did you have sex (vaginal/anal/oral) with any partner when you did not want to because you were afraid of what he might do?", "Within the past year did any partner physically force you to have sex (vaginal/anal/oral) when you did not want to?" and "How many times has this (forced/fear sex/ rape) happened to you in the past 12 months?". Two variables were created using these sexual abuse items (sexual abuse by intimate partner and sexual abuse by non-intimate partner) with responses none vs. some (per above). Furthermore, we created any physical or sexual abuse by an intimate partner and similarly any by a non-intimate partner.

The AUDIT-C scale [5] was adapted and used to indicate severe binge drinking. In addition to the 3 AUDIT questions: "How often do you have a drink containing alcohol?", "How many drinks containing alcohol do you have on a typical day when you are drinking?", and "How often do you have six or more drinks on one occasion?", a question was included on the volume of alcohol per drink (mL). The four items were summed into a score, with a cut-off of 10 used to indicate binge drinking. The Cronbach alpha for the scale was 0.85.

Drug use was measured by asking participants the following questions, "How often have you taken: marijuana (dagga), mandrax, nyaope, cough mixture, ecstasy, methamphetamine (tik), painkillers and rock within the last year?". Cocaine and heroin use were not consistently asked across both studies and therefore were excluded in the model but were included in the descriptive analysis. Responses were coded as "0 never", "1 once", "2 sometimes" and "3 often". Participants scoring ≥1 for any substance question were considered to have used drugs.

## Statistical analysis

Data from both sites were merged, thus the analyses excluded RDS weights and was analysed as a convenience sample. Frequencies and percentages for categorical variables were determined overall, and stratified by site (Soweto and Klerksdorp), by binge drinking, and by drug use. Median and interquartile ranges (IQR) were determined for continuous variables. Chi-square or Fisher's exact tests were used to measure associations between categorical variables whereas the Kruskal-Wallis test was used for continuous variables stratified by binge drinking.

Since drug use data were collected at different times (past year in Soweto and past month in Klerksdorp), data were weighted for analysis on drug use to adjust for bias. Inverse probability weighting (IPW) using propensity scores (ps) was determined. Covariates included in calculating propensity scores were alcohol use, regular clients and age at sex debut. IPW was calculated by the formula $W = \frac{1}{ps}$ if they used drugs and $W = \frac{1}{1-ps}$ if they did not use drugs [52].

Thereafter, three separate weighted Poisson regression models (overall and by site) were used to evaluate risk factors associated with drug use at both univariate and multivariate level. However, risk factors associated with binge drinking were assessed using unweighted Poisson regression modelling as data was collected at the same time points. All variables were included

in the full multivariate model and backward selection method used to select variables for inclusion in the final multivariate models.

Statistical analysis was performed using SAS Enterprise Guide 7.1 (SAS Institute, Cary, NC) and statistical significance was set at p-value $\leq$ 0.05.

## Ethics approval

Ethical approval for both studies was provided by the Human Research Ethics Committee (Medical) of the University of the Witwatersrand, South Africa.

## Results

### Participant characteristics by site

Table 1 details participant characteristics by site; including socio-demographics, HIV status and treatment, mental health, experienced trauma, and substance use.

**Socio-demographics.** There were a total of 664 FSW participants, of whom, 76.51% (508/664) were enrolled in Soweto and 23.49% (156/664) in Klerksdorp. Across both sites, the

**Table 1. Participant characteristics by site.**

| Variable | Overall (n = 664) | Soweto (n = 508) | Klerksdorp (n = 156) | P-Value |
|---|---|---|---|---|
| **Median Age (IQR)** | 30.0 (25.0–36) | 30.0 (25.0–36) | 32.0 (25.0–36) | 0.2887 |
| **Immigration status** | | | | |
| Local (%) | 453/664 (68.22) | 346/508 (68.11) | 107/156 (68.59) | 0.9915 |
| Internal Immigrant (%) | 190/664 (28.61) | 146/508 (28.74) | 44/156 (28.21) | |
| External Immigrant (%) | 21/664 (3.16) | 16/508 (3.15) | 5/156 (3.21) | |
| **Level of education** | | | | |
| No schooling/primary only (%) | 57/664 (8.58) | 37/508 (7.28) | 20/156 (12.82) | 0.0308 |
| Incomplete high school (%) | 450/664 (67.77) | 347/508 (68.31) | 103/156 (66.03) | 0.5938 |
| High/ post school qualification (%) | 157/664 (23.64) | 124/508 (24.41) | 33/156 (21.15) | 0.4026 |
| **Do the people in your home regularly go without food?** | | | | |
| Yes (%) | 466/664 (70.18) | 326/508 (64.17) | 140/156 (89.74) | < .0001 |
| **How old were you when you first had sex?** | | | | |
| Median (IQR) | 17.0 (15.0–18) | 17.0 (15.0–18) | 16.5 (15.0–18) | 0.6007 |
| **Under what circumstances did you first have sex?** | | | | |
| Coercive (%) | 123/664 (18.52) | 110/508 (21.65) | 13/156 (8.33) | 0.0002 |
| **How old were you when you first sold sex?** | | | | |
| Median (IQR) | 24.0 (20.0–30) | 25.0 (20.0–30) | 23.0 (20.0–30) | 0.5847 |
| **HIV Status** | | | | |
| Positive (%) | 361/664 (54.37) | 280/508 (55.12) | 81/156 (51.92) | 0.4834 |
| **Self-reported HIV** | | | | |
| Known Positive | 300/351 (85.47) | 227/271 (83.76) | 73/80 (91.25) | 0.0950 |
| Newly diagnosed | 51/351 (14.53) | 44/271 (16.24) | 7/80 (8.75) | |
| **Are you on treatment?** | | | | |
| Never taken treatment (%) | 94/300 (31.33) | 90/229 (39.30) | 4/71 (5.63) | < .0001 |
| On treatment always (%) | 190/300 (63.33) | 130/229 (56.77) | 60/71 (84.51) | < .0001 |
| Stopped taking treatment (%) | 16/300 (5.33) | 9/229 (3.93) | 7/71 (9.86) | 0.0521 |
| **Self-reported adherence** | | | | |
| Non-adherence (%) | 8/190 (4.21) | 4/130 (3.08) | 4/60 (6.67) | 0.2521 |
| Adherence (%) | 182/190 (95.79) | 126/130 (96.92) | 56/60 (93.33) | |

*(Continued)*

**Table 1.** (Continued)

| Variable | Overall (n = 664) | Soweto (n = 508) | Klerksdorp (n = 156) | P-Value |
|---|---|---|---|---|
| **Depression** | | | | |
| Depressive symptoms (%) | 460/664 (69.28) | 348/508 (68.50) | 112/156 (71.79) | 0.4358 |
| **PTSD** | | | | |
| PTSD symptoms (%) | 212/664 (31.93) | 195/508 (38.39) | 17/156 (10.90) | < .0001 |
| **Childhood trauma** | | | | |
| Some childhood trauma (%) | 590/664 (88.86) | 493/508 (97.05) | 97/156 (62.18) | < .0001 |
| **Physical/sexual abuse from IP within the past year** | | | | |
| Some abuse (%) | 349/664 (52.56) | 288/508 (56.69) | 61/156 (39.10) | 0.0001 |
| **Physical/sexual abuse from non-IP within the past year** | | | | |
| Some abuse (%) | 344/664 (51.81) | 276/508 (54.33) | 68/156 (43.59) | 0.0189 |
| **Binge drinking** | | | | |
| Non-binge drinkers (%) | 291/664 (43.83) | 230/508 (45.28) | 61/156 (39.10) | 0.1741 |
| Binge drinkers (%) | 373/664 (56.17) | 278/508 (54.72) | 95/156 (60.90) | |
| **Within the last year, did you use any drugs?** | | | | |
| No (%) | 469/664 (70.63) | 344/508 (67.72) | 125/156 (80.13) | 0.0029 |
| Yes (%) | 195/664 (29.37) | 164/508 (32.28) | 31/156 (19.87) | |
| **Substance use** | | | | |
| Binge drinking and drug use (%) | 83/664 (12.50) | 59/508 (11.61) | 24/156 (15.38) | 0.2129 |
| Binge drinking or drug use (%) | 402/664 (60.54) | 324/508 (63.78) | 78/156 (50.00) | 0.0021 |
| None (%) | 179/664 (26.96) | 125/508 (24.61) | 54/156 (34.62) | 0.0137 |

median age was 30 years (IQR: 25–36); and 68.2% (453/664) were local residents. Compared to Klerksdorp, Sowetan participants were less likely to have no or primary level education only (7.2% vs. 12.8%; p = 0.0308) and to regularly go without food in their household (64.2% vs 89.7%; p<0.0001).

Respondents' median age at sexual debut was 17 years (IQR: 15–18) and 18.5% (123/664) reported their first sexual encounter as coercive. Participants from Soweto were almost three-fold more likely to be coerced into having sex for the first time (21.7% vs. 8.3%; p = 0.0002) than those from Klerksdorp. The median age of first selling sex across both sites was 24 (IQR: 20–30) years.

**HIV status and treatment.** HIV prevalence at both sites was similar; 55.1% (280/508) in Soweto and 51.9% (81/156) in Klerksdorp (p = 0.4834). Across both sites, of those who tested positive for HIV, 14.5% (51/351) were newly diagnosed. Of the known positives (85.5%, n = 300/351), 63.3% (190/300) self-reported always being on ART, with a higher proportion of participants receiving ART in Klerksdorp compared to Soweto (84.5% vs. 56.8% respectively; p<0.0001). Of the 190 participants always on ART, across both sites, 95.8% (n = 182) self-reported adherence.

**Mental health, experienced trauma, and substance use.** Overall, 69.3% (460/664) and 31.9% (212/664) of participants showed symptoms of depression and PTSD, respectively. About 88.9% (590/664) reported childhood trauma. Though not significant, a slightly higher proportion of FSW in Klerksdorp had depressive symptoms compared to Soweto (71.8% vs. 68.5%; p = 0.4358). Compared to Klerksdorp participants, FSWs from Soweto were fourfold more likely to report PTSD symptoms (38.4% vs. 10.9%; p<0.0001), experienced higher levels of childhood trauma (97.1% vs. 62.2%; p<0.0001), and experienced both physical/sexual abuse by both IPs and non-IPs in the past year (56.7% vs. 39.1%; p = 0.0001; and 54.3% vs. 43.6%; p = 0.0189, respectively).

Of the 664 FSW, 56.2% (373/664) were classified as binge drinkers and 29.4% (195/664) self-reported drug use within the past year (dagga [58.9%, 115/195], mandrax [3.08%, 6/195], nyaope [2.56%, 5/195], cough mixture [29.7%, 58/195], painkillers [27.7%, 54/195], ecstasy [10.3%, 20/195], Tik [4.62%, 9/195] and rock [2.56%, 5/195]). While not significant, a higher percentage of participants from Klerksdorp reported binge drinking compared to Soweto (60.9%, vs 54.7%; p = 0.1741), while significantly more participants from Soweto reported last year drug use (32.3% vs 19.9%; p = 0029). Overall, 12.5% (83/664) of participants reported both binge drinking and drug use with no significant difference by site.

## Participant characteristics by substance use

Table 2 depicts participant characteristics by binge drinking and drug use.

Non-binge drinkers were more likely to have regularly gone without food (74.9% vs. 66.5%; p = 0.0185), and to have experienced physical/sexual abuse from non-IP within the past year (58.4% vs. 46.7%; p = 0.0026).

FSWs who used drugs were more likely to be from Soweto (84.1% vs. 73.4%; p = 0.0001), to have been coerced at first sex (22.6% vs. 16.9%, p = 0.0038), to have PTSD symptoms (43.6%

**Table 2. Participants characteristics by binge drinking and drug use.**

| Variable | Overall | Binge Drinkers | Non-binge Drinkers | P-Value | Some Drug Use | No Drug Use | P-Value |
|---|---|---|---|---|---|---|---|
| **No. enrolled** | 664 (100.00) | 373 (56.17) | 291 (43.83) | - | 195 (29.37) | 469 (70.63) | - |
| **Site** | | | | | | | |
| Klerksdorp (%) | 156/664 (23.49) | 95/373 (25.47) | 61/291 (20.96) | 0.1741 | 31/195 (15.90) | 125/469 (26.65) | 0.0001 |
| Soweto (%) | 508/664 (76.51) | 278/373 (74.53) | 230/291 (79.04) | | 164/195 (84.10) | 344/469 (73.35) | |
| **Median Age (IQR)** | 30.0 (25.0–36) | 30.0 (25.0–36) | 31.0 (25.0–36) | 0.3901 | 31.0 (24.0–36) | 30.0 (25.0–36) | 0.3159 |
| **Do the people in your home regularly go without food?** | | | | | | | |
| Yes (%) | 466/664 (70.18) | 248/373 (66.49) | 218/291 (74.91) | 0.0185 | 138/195 (70.77) | 328/469 (69.94) | 0.6110 |
| **How old were you when you first sold sex?** | | | | | | | |
| Median (IQR) | 24.0 (20.0–30) | 24.0 (20.0–30) | 25.0 (20.0–30) | 0.9016 | 24.0 (19.0–29) | 25.0 (20.0–30) | 0.2100 |
| **HIV Status** | | | | | | | |
| Positive (%) | 361/664 (54.37) | 214/373 (57.37) | 147/291 (50.52) | 0.0784 | 98/195 (50.26) | 263/469 (56.08) | 0.3420 |
| **How old were you when you first had sex?** | | | | | | | |
| Median (IQR) | 17.0 (15.0–18) | 16.0 (15.0–18) | 17.0 (15.0–18) | 0.2503 | 17.0 (15.0–18) | 17.0 (15.0–18) | 0.9999 |
| **Under what circumstances did you first have sex?** | | | | | | | |
| Coercive (%) | 123/664 (18.52) | 69/373 (18.50) | 54/291 (18.56) | 0.9848 | 44/195 (22.56) | 79/469 (16.84) | 0.0038 |
| **Depression** | | | | | | | |
| Depressive symptoms (%) | 460/664 (69.28) | 261/373 (69.97) | 199/291 (68.38) | 0.6598 | 135/195 (69.23) | 325/469 (69.30) | 0.6675 |
| **PTSD** | | | | | | | |
| PTSD symptoms (%) | 212/664 (31.93) | 110/373 (29.49) | 102/291 (35.05) | 0.1272 | 85/195 (43.59) | 127/469 (27.08) | < .0001 |
| **Childhood trauma** | | | | | | | |
| Some childhood trauma (%) | 590/664 (88.86) | 332/373 (89.01) | 258/291 (88.66) | 0.8875 | 180/195 (92.31) | 410/469 (87.42) | 0.0217 |
| **Physical/sexual abuse from IP within the past year** | | | | | | | |
| Some abuse (%) | 349/664 (52.56) | 201/373 (53.89) | 148/291 (50.86) | 0.4381 | 131/195 (67.18) | 218/469 (46.48) | < .0001 |
| **Physical/sexual abuse from non-IP within the past year** | | | | | | | |
| Some abuse (%) | 344/664 (51.81) | 174/373 (46.65) | 170/291 (58.42) | 0.0026 | 133/195 (68.21) | 211/469 (44.99) | < .0001 |
| **Binge drinking** | | | | | | | |
| Binge drinkers (%) | 373/664 (56.17) | - | - | | 83/195 (42.56) | 290/469 (61.83) | 0.8921 |
| **Within the last year, did you use any drugs?** | | | | | | | |
| Yes (%) | 195/664 (29.37) | 83/373 (22.25) | 112/291 (38.49) | 0.8921 | - | - | |

vs. 27.1%; p<0.0001), to have experienced some childhood trauma (92.3% vs. 87.4%, p = 0.0217) and have experienced physically/sexually abuse from their IPs and non-IPs within the past year (67.2% vs. 46.5%, p<0.0001; and 68.2% vs. 45.8%, p<0.0001) compared to those who reported no drug use.

## Risk factors associated with substance use

**Overall risk factors.** Table 3 reports the overall risk factors associated with substance use. Being from Klerksdorp (RR: 1.1957, 95% CI: 1.0277–1.3912), living in a home with regular food provision (RR: 1.2597, 95% CI: 1.1009–1.4413; p = 0.0008), having no or only primary level education (RR: 1.3252, 95% CI: 1.0057–1.746; p = 0.0454) and being HIV positive (RR:

**Table 3. Factors associated with binge drinking and drug use among female sex workers.**

| | Binge drinking | | | | Drug use | | | |
| --- | --- | --- | --- | --- | --- | --- | --- | --- |
| | Univariate | | Multivariate | | Univariate | | Multivariate | |
| Variables | RR 95% (CI) | P-Value | RR 95% (CI) | P-Value | RR 95% (CI) | P-Value | RR 95% (CI) | P-Value |
| **Site** | | | | | | | | |
| Soweto | Ref | - | Ref | - | 1.3146 (1.0781–1.6030) | 0.0069 | 1.1252 (0.8888–1.4246) | 0.3269 |
| Klerksdorp | 1.1128 (0.9592–1.2910) | 0.1585 | **1.1957 (1.0277–1.3912)** | **0.0207** | Ref | - | Ref | - |
| **Age (in years)** | 0.9956 (0.9870–1.0042) | 0.3149 | **0.9815 (0.9675–0.9956)** | **0.0105** | 1.0064 (0.9966–1.0163) | 0.2039 | 1.0030 (0.9929–1.0131) | 0.5632 |
| **Immigration status** | | | | | | | | |
| Local | Ref | - | Ref | - | Ref | - | Ref | - |
| External immigrants | 0.9956 (0.6817–1.4541) | 0.9818 | - | - | 0.7769 (0.4688–1.2876) | 0.3274 | 0.7980 (0.4813–1.3230) | 0.3817 |
| Internal immigrants | 0.9262 (0.7930–1.0818) | 0.3330 | - | - | 0.8640 (0.7229–1.0327) | 0.1082 | 0.8822 (0.7365–1.0567) | 0.1735 |
| **Level of education** | | | | | | | | |
| No/primary only | 1.1805 (0.8986–1.5507) | 0.2333 | **1.3252 (1.0057–1.7461)** | **0.0454** | 1.1630 (0.8751–1.5457) | 0.2981 | - | - |
| Incomplete high school | 1.1917 (0.9979–1.4231) | 0.0528 | 1.1791 (0.9908–1.4032) | 0.0635 | 0.9698 (0.8078–1.1642) | 0.7420 | - | - |
| Matric/post school qualification | Ref | - | Ref | - | Ref | - | Ref | - |
| **Do the people in your home go regularly without food?** | | | | | | | | |
| No | 1.1863 (1.0351–1.3595) | 0.0140 | **1.2597 (1.1009–1.4413)** | **0.0008** | Ref | - | Ref | - |
| Yes | Ref | - | Ref | - | 0.9700 (0.8220–1.1447) | 0.7188 | - | - |
| **How old were you when you first had sex?** | 0.9939 (0.9737–1.0145) | 0.5593 | 0.9848 (0.9643–1.0057) | 0.1529 | 0.9926 (0.9681–1.0177) | 0.5608 | - | - |
| **Circumstance of first sex** | | | | | | | | |
| Non-coercive | Ref | - | Ref | - | Ref | - | Ref | - |
| Coercive | 0.9983 (0.8396–1.1870) | 0.9848 | - | - | 1.2084 (1.0088–1.4475) | 0.0398 | 1.0984 (0.9134–1.3210) | 0.3185 |
| **How old were you when you first sold sex?** | 1.0018 (0.9918–1.0118) | 0.7283 | 1.0168 (1.0002–1.0336) | 0.0472 | 0.9988 (0.9875–1.0102) | 0.8379 | - | - |
| **HIV status** | | | | | | | | |
| Negative | Ref | - | Ref | - | Ref | - | Ref | - |

*(Continued)*

**Table 3.** (Continued)

| Variables | Binge drinking | | | | Drug use | | | |
|---|---|---|---|---|---|---|---|---|
| | Univariate | | Multivariate | | Univariate | | Multivariate | |
| | RR 95% (CI) | P-Value | RR 95% (CI) | P-Value | RR 95% (CI) | P-Value | RR 95% (CI) | P-Value |
| Positive | 1.1297 (0.9850–1.2956) | 0.0813 | **1.1678 (1.0227–1.3334)** | **0.0219** | 0.9490 (0.8148–1.1054) | 0.5012 | - | - |
| **PTSD** | | | | | | | | |
| No PTSD symptoms | Ref | - | Ref | - | Ref | - | Ref | - |
| PTSD symptoms | 0.8917 (0.7665–1.0375) | 0.1379 | - | - | 1.3217 (1.1311–1.5445) | 0.0004 | **1.1803 (1.0025–1.3895)** | **0.0465** |
| **Childhood trauma** | | | | | | | | |
| No childhood trauma | Ref | - | Ref | - | Ref | - | Ref | - |
| Some Childhood trauma | 1.0156 (0.8180–1.2610) | 0.8883 | - | - | 1.2528 (0.9553–1.6430) | 0.1032 | 0.9491 (0.6913–1.3029) | 0.7464 |
| **Physical/sexual abuse from IP within the past year** | | | | | | | | |
| No abuse | Ref | - | Ref | - | Ref | - | Ref | - |
| Some abuse | 1.0548 (0.9215–1.2073) | 0.4392 | - | - | 1.5515 (1.3202–1.8232) | < .0001 | **1.3648 (1.1522–1.6167)** | **0.0003** |
| **Physical/sexual abuse from non-IP within the past year** | | | | | | | | |
| No abuse | Ref | - | Ref | - | Ref | - | Ref | - |
| Some abuse | 0.8134 (0.7107–0.9309) | 0.0027 | **0.8002 (0.6993–0.9156)** | **0.0012** | 1.5369 (1.3096–1.8036) | < .0001 | **1.3910 (1.1793–1.6407)** | **< .0001** |

1.1678, 95% CI: 1.0227–1.3334; p = 0.0219) were associated with a higher risk for binge drinking. However, an increase in age per year (RR: 0.9815, 95% CI: 0.9675–0.9956; P = 0.0105), and physical/sexual abuse from non-IPs within the past year (RR: 0.8002, 95% CI: 0.6993–0.9156; P = 0.0012) were associated with lower risk of binge drinking.

Overall, having PTSD symptoms (RR: 1.1803, 95% CI: 1.0025–1.3895; p = 0.0465) and past year physical/sexual abuse from either IPs (RR: 1.3648, 95% CI: 1.1522–1.6167; p = 0.0003) or non-IPs (RR: 1.3910, 95% CI: 1.1793–1.6407; p<0.0001) were associated with higher risk of drug use.

## Site-specific risk factors associated with substance use

Table 4 reports the risk factors for substance use by site.

**Soweto-specific risk factors associated with binge drinking and drug use.** In Soweto, FSWs living in a home with regular food provision (RR: 1.2795, 95% CI: 1.1025–1.4859; p = 0.0012) and being HIV positive (RR: 1.2030, 95% CI: 1.0290–1.4065; p = 0.0204) had an increased risk of binge drinking, whereas those who had experienced physical/sexual abuse from non-IP within the past year (RR: 0.7843, 95% CI: 0.6739–0.9128; p = 0.0017) had a lower risk of binge drinking.

Higher risk of drug use was associated with past year of physical/sexual abuse from both IPs (RR: 1.4563, 95% CI: 1.1997–1.7678; p = 0.0001) and non-IPs (RR: 1.4606, 95% CI: 1.2115–1.7611; p<0.0001). Being an internal immigrant was associated with lower risk of drug use (RR: 0.7888, 95% CI: 0.6420–0.9690; p = 0.0239).

**Klerksdorp-specific risk factors associated with binge drinking and drug use.** In Klerksdorp, physical/sexual abuse from an IP within the past year was associated with an increased risk of binge drinking (RR: 1.4298, 95% CI: 1.1232–1.8201; p = 0.0037), while an

**Table 4. Factors associated with binge drinking and drug use among female sex workers in Soweto and Klerksdorp.**

| | Binge drinking | | | | Drug use | | | |
| --- | --- | --- | --- | --- | --- | --- | --- | --- |
| | Univariate | | Multivariate | | Univariate | | Multivariate | |
| Variables | RR 95% (CI) | P-Value | RR 95% (CI) | P-Value | RR 95% (CI) | P-Value | RR 95% (CI) | P-Value |
| **SOWETO** | | | | | | | | |
| **Age (in years)** | 1.0036 (0.9933–1.0140) | 0.4948 | | | 1.0135 (1.0027–1.0245) | 0.0145 | 1.0091 (0.9982–1.0200) | 0.1013 |
| **Immigration status** | | | | | | | | |
| Local | Ref | - | Ref | - | Ref | - | Ref | - |
| External immigrants | 1.1689 (0.7898–1.7300) | 0.4352 | - | - | 0.8912 (0.5370–1.4792) | 0.6560 | 0.9471 (0.5701–1.5734) | 0.8338 |
| Internal immigrants | 1.0632 (0.8951–1.2630) | 0.4851 | - | - | 0.7796 (0.6351–0.9570) | 0.0173 | **0.7888 (0.6420–0.9690)** | **0.0239** |
| **Do the people in your home go regularly without food?** | | | | | | | | |
| No | 1.2637 (1.0821–1.4759) | 0.0031 | **1.2795 (1.1025–1.4849)** | **0.0012** | Ref | - | Ref | - |
| Yes | Ref | - | Ref | - | 1.0499 (0.8790–1.2540) | 0.5914 | - | - |
| **How old were you when you first had sex?** | 0.9973 (0.9723–1.0230) | 0.8376 | - | - | 1.0030 (0.9755–1.0313) | 0.8323 | - | - |
| **HIV status** | | | | | | | | |
| Negative | Ref | - | Ref | - | Ref | - | Ref | - |
| Positive | 1.1714 (0.9953–1.3787) | 0.0570 | **1.2030 (1.0290–1.4065)** | **0.0204** | 0.9129 (0.7715–1.0803) | 0.2887 | - | - |
| **PTSD** | | | | | | | | |
| No PTSD symptoms | Ref | - | Ref | - | Ref | - | Ref | - |
| PTSD symptoms | 0.8878 (0.7503–1.0504) | 0.1653 | - | - | 1.2589 (1.0630–1.4909) | 0.0076 | 1.1572 (0.9739–1.3750) | 0.0970 |
| **Childhood trauma** | | | | | | | | |
| No childhood trauma | Ref | - | Ref | - | Ref | - | Ref | - |
| Some Childhood trauma | 1.3793 (0.7384–2.5766) | 0.3131 | - | - | 1.8071 (0.8578–3.8071) | 0.1196 | 1.4149 (0.6654–3.0084) | 0.3672 |
| **Physical/sexual abuse from IP within the past year** | | | | | | | | |
| No abuse | Ref | - | Ref | - | Ref | - | Ref | - |
| Some abuse | 0.9350 (0.7982–1.0953) | 0.4052 | - | - | 1.6427 (1.3621–1.9811) | < .0001 | **1.4563 (1.1997–1.7678)** | **0.0001** |
| **Physical/sexual abuse from non-IP within the past year** | | | | | | | | |
| No abuse | Ref | - | Ref | - | Ref | - | Ref | - |
| Some abuse | 0.8051 (0.6877–0.9424) | 0.0070 | **0.7843 (0.6739–0.9128)** | **0.0017** | 1.6267 (1.3557–1.9519) | < .0001 | **1.4606 (1.2115–1.7611)** | **< .0001** |
| **KLERKSDORP** | | | | | | | | |
| **Age (in years)** | 0.9751 (0.9593–0.9912) | 0.0026 | 0.9853 (0.9688–1.0021) | 0.0862 | 0.9720 (0.9481–0.9964) | 0.0249 | **0.9733 (0.9494–0.9977)** | **0.0324** |
| **Immigration status** | | | | | | | | |
| Local | Ref | - | Ref | - | - | - | - | - |
| External immigrants | 0.5707 (0.1937–1.6815) | 0.3090 | 0.6615 (0.2247–1.9472) | 0.4531 | - | - | - | - |
| Internal immigrants | 0.5836 (0.4007–0.8501) | 0.0050 | 0.8101 (0.5211–1.2592) | 0.3493 | - | - | - | - |
| **Level of education** | | | | | | | | |

(*Continued*)

**Table 4.** (Continued)

| | Binge drinking | | | | Drug use | | | |
|---|---|---|---|---|---|---|---|---|
| | Univariate | | Multivariate | | Univariate | | Multivariate | |
| Variables | RR 95% (CI) | P-Value | RR 95% (CI) | P-Value | RR 95% (CI) | P-Value | RR 95% (CI) | P-Value |
| How old were you when you first had sex? | 0.9863 (0.9565–1.0170) | 0.3776 | - | - | 0.9508 (0.8999–1.0044) | 0.0716 | 0.9533 (0.9004–1.0094) | 0.1009 |
| Physical/sexual abuse from IP within the past year | | | | | | | | |
| No abuse | Ref | - | Ref | - | | | | |
| Some abuse | 1.5905 (1.2492–2.0251) | 0.0002 | **1.4298 (1.1232–1.8201)** | **0.0037** | 1.0985 (0.7630–1.5816) | 0.6133 | - | - |

increase in age was associated with lower risk of drug use (RR: 0.9733, 95% CI: 0.9494–0.9977; p = 0.0324).

## Discussion

Our analysis of 664 FSWs in two provinces of South Africa shows a high prevalence of substance use, with over half of the participants meeting criteria for binge drinking and more than a quarter engaging in drug use. Overall, living in a home with regular food and being HIV positive were associated with a higher risk of binge drinking. Having symptoms suggestive of PTSD and past year physical/sexual abuse from either IPs or non-IPs were associated with a higher risk of drug use.

Substance use is a growing public health concern, especially given the increasing rates of drug use in South Africa [17, 53]. As a key vulnerable population already underutilizing public health services, FSWs are at an added disadvantage. The prevalence of binge drinking (56.2%) reported in amongst FSWs in our studies is almost tenfold that of women in South Africa (6.4%) [54]. The overall prevalence of drug use in the past year (29.4%) is within the range given for FSWs in various South African cities (24.5–39.8%), even though it is an underestimate given the exclusion of cocaine and heroin [6]. Furthermore, our findings demonstrate that the risk factors for alcohol and drug use differ per site, thus highlighting the need for site-specific nuances to targeted interventions serving FSWs. This is the first study to describe FSWs in Klerksdorp, making these findings even more pertinent to an area with a high HIV burden [40] and with limited educational and employment opportunities for women [55, 56].

Drug use was significantly higher in Soweto compared to Klerksdorp. This could be a result of geographic scope and contextual factors, with Soweto being a larger area within a more urban setting and thus possibly having better access to drugs [25]. Critically, our study shows an inverse relationship between drug use and binge drinking across both sites, which could suggest that under certain social circumstances, FSWs may choose between alcohol or drugs, rather than both. It is concerning that South Africa has such a paucity in treatment facilities for substance use [57], with limited vertical programmes for FSWs to access support while continuing to earn a living. There is an urgent need to create integrated substance use programmes that support FSWs in ways appropriate to their needs and contextual dynamics.

Our results demonstrate a significant burden of self-reported mental health issues. Almost three-quarters of FSW reported suffering from symptoms of major depression. This is almost seven times higher than in the general South African population [16]. Compared to Klerksdorp, Soweto showed a higher proportion of PTSD symptoms, which is associated with high rates of childhood trauma and physical/sexual violence [58]. Our study shows that PTSD symptoms are associated with a higher risk of drug use. These findings may explain the higher

prevalence of drug use in Soweto. Interestingly, our analysis of depression did not present as a strong risk factor for substance use, even though previous research has shown substance use to be a key risk factor for mental health problems among FSWs in low- and middle-income countries (LMICs) [59].

The findings of our study corroborate previous work showing the high prevalence of sexual and/or physical violence exposure amongst FSWs [6, 25]. This is of particular importance in Klerksdorp, where there is no previous research describing violence exposure for FSWs. In comparison to Soweto, Klerksdorp FSWs experienced less childhood trauma and were less likely to describe their first sexual experience as coercive. They were also significantly less likely to report exposure to past year physical and/or sexual violence–a major risk factor for substance use, which our study also reports [6]. It is unsurprising then that we find lower levels of substance use (both alcohol and drug use) in Klerksdorp, where violence exposure was lower. Meanwhile, variables such as food security and a positive HIV status increased the risk of binge drinking. This is in contrast to literature that suggests that one of the most common structural drivers of substance use in occupational groups at high risk of HIV in South Africa is widespread poverty [60]. Food insecurity and substance use have been shown to be highly correlated for men but not women [61]. The latter variable is unsurprising given the demonstrated association between substance use and medication non-adherence, and the explanation that substance use can serve as a coping mechanism for dealing with a chronic illness [62]. The risk factor of a positive HIV status for binge drinking is notable because heavy drinking can lead to high-risk sexual behaviours, increased potential for HIV transmission, sub-optimal ART adherence, and other health complications including liver disease [63].

Our study had several limitations. The smaller sample size in Klerksdorp likely impacts the precision of the findings. The data was collected at differing time periods at each site, and changes in context could account for differences in findings. This is of particular importance for ART uptake, which due to a policy change in late 2016, improved access for HIV positive FSW. Self-reported ART-use could also have been impacted by social desirability bias. The AUDIT-C short measure is possibly inadequate to assess chronic binge drinking. Drug use was measured differently across the two sites: Klerksdorp asked about drug use in the past month and Soweto asked about drug use in the past year. As such the variable explored drug use "within the last year". Cocaine and heroin use was not measured in Soweto and therefore was excluded from the model. Additionally, we do not have longitudinal data on substance use.

## Conclusions and recommendations

Our research shows the high prevalence of substance use by FSWs in two locations in South Africa, which suggests either the lack of awareness of or access to effective interventions. Urgent interventions are required to address this concern. The high rates of physical and sexual abuse throughout the lives of FSW and ill mental health is noteworthy, given the known links between/among trauma, ill mental health and substance use. Interventions for physical and sexual abuse are required not only at the individual-level, but also the community-level to include male intimate partners and non-intimate partners who perpetrate violence, and structural-level to address legislative frameworks which drive vulnerability amongst FSWs while allowing male partners to act with impunity. Given the trauma exposure of this population, it is unsurprising that such high rates of substance use are found. Interventions targeted at drivers of mental illness are needed, as well as those geared towards specifically supporting women suffering from mental illness.

There is a current paucity of knowledge on substance use and sex worker dynamics, which will likely hinder efforts to develop effective long-term substance use, mental health, and

gender-based violence-related interventions. Our findings highlight site-specific contextual dynamics driving substance use and can be used to better tailor interventions for FSWs in South Africa, where contextual dynamics vary widely across the country. Broader research is needed to better understand the site-specific risk factors for alcohol and/or drug use in populations with an increased risk of substance use due to occupation-related factors.

## Supporting information

**S1 Questionnaire. Klerksdorp version of complete study questionnaire.**
(PDF)

**S1 Table. Questionnaire tools.**
(PDF)

## Author Contributions

**Conceptualization:** Ebrahim Variava, Neil Martinson, Steffanie A. Strathdee, Jenny Coetzee, Minja Milovanovic.

**Data curation:** Jenny Coetzee, Minja Milovanovic.

**Formal analysis:** Khuthadzo Hlongwane, Kennedy Otwombe.

**Methodology:** Jenny Coetzee.

**Resources:** Jenny Coetzee.

**Writing – original draft:** Ellis Jaewon Yeo, Khuthadzo Hlongwane, Minja Milovanovic.

**Writing – review & editing:** Ellis Jaewon Yeo, Kathryn L. Hopkins, Minja Milovanovic.

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
