## [Decision Letter · Decision Letter 0]

31 Aug 2021

PONE-D-21-22797

Key Risk Factors for Substance Use among Female Sex Workers in Soweto and Klerksdorp, South Africa: A cross-sectional study

PLOS ONE

Dear Dr. Yeo,

Thank you for submitting your manuscript to PLOS ONE. After careful consideration, we feel that it has merit but does not fully meet PLOS ONE’s publication criteria as it currently stands. Therefore, we invite you to submit a revised version of the manuscript that addresses the points raised during the review process.

We look forward to receiving your revised manuscript.

Kind regards,

Yukiko Washio, Ph.D.

Academic Editor

PLOS ONE

Journal Requirements:

2. Please include additional information regarding the survey or questionnaire used in the study and ensure that you have provided sufficient details that others could replicate the analyses. For instance, if you developed a questionnaire as part of this study and it is not under a copyright more restrictive than CC-BY, please include a copy, in both the original language and English, as Supporting Information. Moreover, please include more details on how the questionnaire was pre-tested, and whether it was validated. 

Reviewers' comments:

Reviewer's Responses to Questions

**Comments to the Author**

1. Is the manuscript technically sound, and do the data support the conclusions?

Reviewer #1: Yes

Reviewer #2: Partly

2. Has the statistical analysis been performed appropriately and rigorously? 

Reviewer #1: Yes

Reviewer #2: No

3. Have the authors made all data underlying the findings in their manuscript fully available?

Reviewer #1: Yes

Reviewer #2: Yes

4. Is the manuscript presented in an intelligible fashion and written in standard English?

Reviewer #1: Yes

Reviewer #2: Yes

5. Review Comments to the Author

Reviewer #1: Thank you for opportunity to review. Apologies for brevity. Your reviewer is unwell with Covid but I wanted to get this back so the authors could address speedily.

The subject matter explored is highly relevant and speaks to understanding and meeting the needs of a key generally marginalised population. Congrats!

The literature review is appropriate. There are however a few brief points of reflection.

In the abstract:

1. At line 37: Include a note about which survey? was it developed for this study? Evaluating what exactly? Validated?

2. At line 40, with reference to the text “key risk factors of alcohol and drug use” it might assist if this was clearer on whether this referred to Risks for or risks resulting from or all risks?

In the introduction

3. At lines 58 and 59, the authors might perhaps consider using more current references. For line 58 The authors need to cite more current publications such as the most recent World Drug Report, which will have been published in the period since this manuscript was probably submitted.

4. Line 60 refers to a diagnosis of drug use. Were the authors intending to refer to a diagnosis of substance use disorder? Can drug use be “diagnosed”?

5. The final sentence of the first paragraph at lines 62 and 63, This statement is not supported by the provided stats. The sentence should perhaps read "...vary widely and this is likely a consequence of cultural and other environment factors" This statement would itself need a reference.

6. At line 86, Should there be centres specifically dedicated to FSW? Would this not further marginalize and stigmatize this population?

7. At line 87, the text “access to the few treatment centres is limited” refers. In what way is access limited? Some brief reflection on this might assist.

8. At line 89, there is reference to competing financial priority while in treatment. In what way? Could the authors provide a brief text for clarity in the manuscript.

9. At line 112 there is an appropriate refence to relationship to viral suppression consequent upon poor adherence. The links need to be stated more explicitly here and in terms of reduces access to health care in the next sentence, to clearly set the scene.

10. At line 117, The correct note about substance use being currently infrequently studied, as well needs to be more explicit. What about the substance use is infrequently studied? Is it substance use patterns? Is it substance use in general. I know what the authors are referring to but the writing needs to be more explicit for any reader.

In the Methods Section

11. At line 123, Is this manuscript reporting on two studies, or is it reporting on the findings of data collected at two sites at different times? Do these studies have separate Ethical clearance or is the newer sample an extension of the original approval?

12. At line 124, Brief description to link to the next sentence, RDS methodology, where an initial group of respondents assembled via convenience sampling are asked to invite additional participants from within their network" or something along those lines...

13. At line 133, the statement indicating the population size of Soweto needs a reference.

14. At line 144: As I am familiar with the implementing partner landscape in South Africa, I know what this means, but a reader from elsewhere might not be clear. This sentence should more explicitly clarify what these programmes are, what they do so as to clearly link the reader to why it is that they would have access to and the trust of FSW's

15. At line 157, Per above , perhaps some brief context into what the sex worker programme is.

16. At line 168 regarding data collection, what is meant by privately and collaboratively? Could this be made clearer in the text of the article?

17. In general the data collection process is not clear. The authors should seek to describe this in a way that is easy to follow and replicate.

18. At line 174 the authors refer to RDS assumptions and how these were managed. Which assumptions specifically?

19. At line 181 the following text refers “A detailed description of all measures used has been included in other papers” Please still at least list the measures or at least mention they are detailed below. (Having subsequently read further) Seeing as they are detailed below I'm not sure this sentence is essential)

20. At line 185, could the authors please clarify “cognitive interviews”? This can be confusing, particularly for this population as this is so often used in reference to law enforcement procedures.

21. The authors refer to physical and sexual use by partner throughout the manuscript. Is this correct or is it meant to refer to “abuse”. If “use is correct, please define?

22. My initial feeling was that the tables were possibly cumbersome, but they are in fact very handy, and I hope they can be retained in the final publication.

The authors have done an excellent job of characterising this population at these sites and eliciting their risk profiles for harmful substance use.

This material is not only suitable fir publication but will add great value in the consideration of responsive service provision for this key population. It should certainly be considered for publication with these minor revisions.

Authors have largely adhered to the STROBE recommendations, which apply to this type of study. They should refer more explicitly to the ways in which they have done so, and this might assist in addressing some minor gaps. For example, with this population is is especially important to consider ethical issues and speak to how potential bias has been addressed.

Per my input above, the procedures of the study could be described somewhat more explicitly so as to assist with replicability.

The manuscript is well organized but some clarifications and suggested above might assist in making the material more accessible to a broader readership.

This manuscript in my view does not contain an NIH-defined experiment of Dual Use Concern

Reviewer #2: Overall, really interesting paper with really interesting findings. There is just one piece that concerns me is drug use in the past month for Klerksdorp population and drug use in the past year for the Soweto population. See my last comment for suggestions on how to address this. All other comments are minor revisions.

Line 116-117 The intro section is full of strong statistics about SU in FSW populations. It is a bit of a contradiction to say something has been infrequently studied, when you have presented several articles to support SU in FSW populations (Line 65-68, 70, 72,77). The sentence should read “despite the high prevalence of trauma and mental health, gaps exist when focusing on binge drinking and drug use (although I do think you provided evidence that there is no gap when focused on drug use). The focus of this study is to describe the prevalence of self-reported binge drinking, and associated risk factors” or something similar. I think just a simple reivision will address this

Line 124 “enroll” Small spelling error

Line 126-127 “The full methodology from the Soweto study has been extensively described previously (7)” Language could be a better. For example: “The methods from the Soweto study has been throughly described in previous publications (7)”  

Keep language consistent. Sometimes you utilize “FSW” and sometimes “woman.” For consistency, choose one! (End of line 136).

 Line 137 “informal sex work was also paid for in beer” Do you have a citation for this?

Line 168 “Post enrollment” Another small spelling error.

Line 411-413 This is interesting and could also be a focus of future research. In what circumstances are FSW using both?

Line 450-452 The focus on HIV and ART feels a little out of place. I understand the big picture relationship, but how does this relate to the objective of the analysis which was to determine the prevalence and associated risks of binge drinking and drug use?  Other limitations: cross sectional design.

Line 455: Because the Kleksdorp population was only asked about past month drug use, was this adjusted to reflect what the past year use would be? This assumes that every month has similar patterns of drug use, which I think is a feasible assumption. This could also explain why drug use was higher in Soweto compared to Kerksdorp. I think this is a glaring problem that needs to be adjudicated in either the methods to explain how you adjusted for this discrepancy or in the results, where you use the one month report to predict one years use, using the assumption the the report of one month's use is reflective of use in all the other months. This may change your results.

6. PLOS authors have the option to publish the peer review history of their article (what does this mean?). If published, this will include your full peer review and any attached files.

Reviewer #1: No

Reviewer #2: No

---

## [Author Response · Author response to Decision Letter 0]

16 Nov 2021

To address the additional requirements noted in the Decision Letter:

1. Manuscript has been edited to meet PLOS ONE's style requirements.

2. The questionnaire used for the Soweto and Klerksdorp study was developed by merging existing and validated tools such as: 10-item CES-D short scale for depression, PTSD-8 scale, The AUDIT-C scale for alcohol use, The Childhood Trauma Questionnaire and the WHO violence against women (adapted for female sex workers). All of these questionnaires have been referenced in the manuscript.

3. We have not uploaded the data onto a data repository and now included the below sentences under the Data Availability Statement: Female sex workers are a highly vulnerable key population due to violence, stigma and the criminalised nature of the work. For this reason it is imperative that data collected from studies with female sex workers follow ethical considerations and account for the sensitive nature of the information. Therefore, datasets will be available upon request by contacting info@phru.co.za. 

Please see the "Response to Reviewers" document for responses that correspond to reviewer comments. (Responses have been copied below but will be better understood when viewing the comments side by side in the document)

Dear Reviewer thank you for your comment. Please find our specific responses below. 

Thank you for your comment. A note has been included in the abstract highlighting that the survey was developed for the study and the information collected included socio-demographic, mental-ill health, violence and HIV status. 

This sentence has been revised to “risk factors for alcohol and drug use.”

The reference has been updated to a more recent 2018 WHO report. 

The sentence has been revised to “diagnosis of substance use disorder.”

The sentence has been revised. The statement uses the same reference as the preceding sentence. 

This is an important point. Through our work with FSWs we have learnt that by having a dedicated sex workers only space, that provides targeted health services in a safe and trusting environment, we are able to increase the uptake of health services and treatment adherence while simultaneously providing psycho-social support. However, it is important to ensure that dedicated spaces are run by personnel who have been sensitized to working with key populations to mitigate stigmatization and marginalization. Due to the nature of sex work, FSW risk being marginalized and stigmatized when accessing services from general facilities and substance use could potentially add to this stigma. We therefore, believe that there should be centres specifically dedicated to FSW because they face unique challenges and risk factors and experience a high prevalence of substance use disorder. We believe that the benefits of a FSW only centre would minimize the risk of stigmatization and create a space where FSWs can seek treatment specific to their needs. 

Thank you for your comment. This statement has been clarified such that access is limited by limited availability of space and services. 

The “competing financial priority” refers to the fact that FSW cannot work and earn an income while they are in treatment. The sentence has been revised for clarification. 

Thank you for your comment. The statement has been amended for clarity and the sentence referring to ‘access to healthcare’ has been moved to paragraph three. 

The sentence has been revised to “patterns of substance use.”

This manuscript is reporting on findings of data collected from two studies. The studies were conducted at different sites but followed the same methodology. The two studies have separate ethical clearance certificates. This has been made clearer in the manuscript. 

The phrase has been adapted to provide a stronger link to the next sentence. 

A reference to the 2011 census has been added. 

More information about the sex worker programmes have been added. 

More information about the sex worker programmes have been added. 

Thank you for the comment, the sentence has been revised for clarity. 

The data collection process has been amended for clarity and replicability. 

The RDS assumptions made include: that the population being recruited must know one another as FSW, participants recruited have similar characteristics, must be networked and could accurately estimate their network size and peer to peer recruitment is random. A reference has been added in the manuscript that refers to the RDS assumptions. 

Thank you for the comment, the sentence has been deleted. 

Thank you for your comment, cognitive interviews were conducted to assess the understandability and appropriateness of the questionnaire. This has been described in the manuscript.

Thank you for the comment. This was an error and all instances of “physical/sexual use” has been corrected to “physical/sexual abuse.”

Thank you for the comment. We will keep the tables in the final publication. 

Thank you for these final remarks, we have reviewed the STROBE recommendations and hope that in addressing the reviewers comments we have managed to fill any noted gaps especially around ethical issues and bias. 

Dear reviewer, thank you for your comments. Please find our specific responses below.

We revised the sentence to incorporate both your and Reviewer 1’s feedback. The sentence now reads “Despite the high prevalence of trauma and mental health concerns reported by FSWs in South Africa, gaps exist when focusing on patterns of substance use for this population.” 

The spelling error has been corrected. 

The sentence has been revised to “the methodology from the Soweto study has been thoroughly described in previous publications.”

The end of line 136 has been switched to “FSW.”

This was an error. Citation 24 was moved to encompass the entire sentence which it refers to. 

The spelling error has been corrected. 

Thank you for your comment. While we do agree that understanding the circumstances for both binge drinking and drug use would be an interesting topic for future research our numbers were too small to compare for this manuscript especially as it would need to be split by site. 

Thank you for your comment. As noted in the introduction, previous studies have found that substance use is linked to HIV and ART adherence. Additionally, HIV status has been associated with substance use with our analysis finding that Hiv status increased risk of binge drinking. As FSW have a higher HIV prevalence than the general population we believe it is important to include HIV and ART when determining the patterns and risk factors for substance use amongst this population. 

We have noted in the limitations section that we do not have longitudinal data on substance use. 

Thank you to the reviewer for the comment. We decided to use Inverse Probability Weighting using propensity score to adjust for the time difference in both site. We have explained the process in the method section and updated the results to account for the adjustment done.

---

## [Decision Letter · Decision Letter 1]

13 Dec 2021

Key risk factors for substance use among female sex workers in Soweto and Klerksdorp, South Africa: A cross-sectional study

PONE-D-21-22797R1

Dear Dr. Yeo,

We’re pleased to inform you that your manuscript has been judged scientifically suitable for publication and will be formally accepted for publication once it meets all outstanding technical requirements.

Kind regards,

Yukiko Washio, Ph.D.

Academic Editor

PLOS ONE

Additional Editor Comments (optional):

Reviewers' comments:

Reviewer's Responses to Questions

**Comments to the Author**

1. If the authors have adequately addressed your comments raised in a previous round of review and you feel that this manuscript is now acceptable for publication, you may indicate that here to bypass the “Comments to the Author” section, enter your conflict of interest statement in the “Confidential to Editor” section, and submit your "Accept" recommendation.

Reviewer #1: All comments have been addressed

Reviewer #2: All comments have been addressed

2. Is the manuscript technically sound, and do the data support the conclusions?

Reviewer #1: Yes

Reviewer #2: Yes

3. Has the statistical analysis been performed appropriately and rigorously? 

Reviewer #1: Yes

Reviewer #2: Yes

4. Have the authors made all data underlying the findings in their manuscript fully available?

Reviewer #1: No

Reviewer #2: No

5. Is the manuscript presented in an intelligible fashion and written in standard English?

Reviewer #1: Yes

Reviewer #2: Yes

6. Review Comments to the Author

Reviewer #1: Initial comments have been sufficiently addressed and manuscript reads strongly and clear. A single additional minor recommendation might be where it is said that FSW'w were used to identify additional participants. Could the authors consider a word or phrase to say this more sensitively for what is a vulnerable population?

Reviewer #2: As I mentioned before, I really enjoyed reading this paper. Great job with the revisions. Best of luck!

7. PLOS authors have the option to publish the peer review history of their article (what does this mean?). If published, this will include your full peer review and any attached files.

Reviewer #1: No

Reviewer #2: No

---

## [Editor Report · Acceptance letter]

11 Jan 2022

PONE-D-21-22797R1 

Key risk factors for substance use among female sex workers in Soweto and Klerksdorp, South Africa: A cross-sectional study 

Dear Dr. Yeo:

I'm pleased to inform you that your manuscript has been deemed suitable for publication in PLOS ONE. Congratulations! Your manuscript is now with our production department. 

Kind regards, 

on behalf of

Dr. Yukiko Washio 

Academic Editor

PLOS ONE